# Prevalence and risk factors of asymptomatic bacteriuria in community-dwelling Korean adults

**Hyun Lee Ko**  ¶, **Soojin Lee**◎, **Dha Woon Im**◎, **Sung Woo Lee***

Division of Nephrology, Department of Internal Medicine, Uijeongbu Eulji University Medical Center, Uijeongbu-si, Gyeonggi-do, Republic of Korea

¶ First author
◎ These authors contributed equally to this work
* neplsw@gmail.com

## Abstract

The clinical characteristics of asymptomatic bacteriuria (ASB) in community-dwelling Asian adults remain underexplored. We aimed to identify the potential risk factors for ASB in an Asian population. We conducted a cross-sectional analysis of 8,508 participants. Urinary bacteria were quantified using urine flow cytometry and semi-quantitatively reported on a scale from negative to 3+. For this epidemiologic analysis, bacteriuria was operationally defined as a flow cytometry result of ≥1+, rather than by culture-based criteria. ASB was defined as bacteriuria accompanied by a self-reported negative answer to the question: "Have you recently experienced any sudden and frequent need to urinate?" The overall prevalence of ASB was 4.6%, with rates of 5.9% in women and 1.0% in men under 50, and 9.5% in women and 1.6% in men over 50. Older age, women, and diabetes were associated with increased odds of having ASB, whereas high physical activity and serum albumin levels were associated with decreased odds. Regardless of age, diabetes status, and physical activity and serum albumin levels, women showed higher odds of ASB than did men. Men and women with high physical activity and serum albumin levels showed lower odds of ASB than those without high levels. Age, sex, diabetes, physical activity, and serum albumin levels were independently associated with ASB among community-dwelling Korean adults. Future studies need to confirm the potential beneficial association of high physical activity and serum albumin levels with ASB in men and women.

## Introduction

Asymptomatic bacteriuria (ASB) is diagnosed when an excessive quantity of bacteria is identified in an uncontaminated culture from a urine sample without the patient showing any symptoms of a urinary tract infection (UTI) [1]. Although ASB is an independent risk factor for future UTIs [2], routine antibiotic use is not recommended because it does not significantly prevent UTIs or improve the quality of life of people

**Data availability statement:** The data underlying the results of this study are available from the Korean Genome and Epidemiology Study (KoGES), conducted by the Korea Disease Control and Prevention Agency (KDCA), Republic of Korea. Due to ethical and legal restrictions, the data are not publicly available. However, the data can be accessed by qualified researchers through a formal application process administered by the KoGES Data Access Committee of the KDCA. Requests for data access should be submitted via the KoGES data access portal (https://coda.nih.go.kr/frt/index.do). The authors did not have any special access privileges. Any researcher who meets the criteria for access may obtain the data in the same manner.

**Funding:** EMBRI Grants 2024-EMBRIUI-0004 from the Eulji University. The funders had no role in study design, data collection and analysis, decision to publish, or preparation of the manuscript.

**Competing interests:** The authors have declared that no competing interests exist.

with ASB [3]. This indicates a complex relationship between ASB and symptomatic UTIs. Despite the clinical significance of ASB, its characteristics have not been fully evaluated, especially in large-scale community-dwelling populations, likely owing to the difficulty in obtaining one or more urine culture results.

Although a urine culture is the gold standard for diagnosing bacteriuria, it is time-consuming and laborious, requiring trained technicians and biological contamination-free spaces that may not be available in community hospitals [4]. Urine sediment analysis via microscopy can be used as a screening tool for bacteriuria; however, it requires specialized training of laboratory staff and is prone to a high level of variability between different operators. Recent technological advancements have enabled the rapid counting of bacteria via flow cytometry [5]. This approach produces results that correlate well with manual counting under a microscope [6]. Additionally, flow cytometry has good sensitivity and specificity for positive urine culture results [4]. Because flow cytometers involve the use of uncentrifuged urine, bacteriuria can be easily detected for large-scale, population-based cohort studies.

Older age, female sex, pregnancy, and diabetes have been suggested as risk factors for developing ASB in adults [1]. However, to the best of our knowledge, no large-scale studies exist on community-dwelling adult populations in which ASB is characterized or socioeconomic status is considered. Therefore, this study aimed to identify potential risk factors associated with ASB in an Asian population using data from the Ansung-Ansan cohort of the Korean Genome Epidemiology Study (KoGES).

## Materials and methods

### Ethics of experimentation statement

The study protocol adhered to the principles of the Declaration of Helsinki and was approved by the Ethics Committee of Uijeongbu Eulji Medical Center, Eulji University (approval no. 2022-12-001). Although this study retrospectively analyzed anonymized data extracted from KoGES, written informed consent had been obtained from all participants at the time of data collection. The data were accessed for research purposes on 7 January 2023. All data were fully anonymized before access, and at no point during or after data collection did the authors have access to identifiable personal information.

### Participants

A total of 10,030 individuals aged 40–69 years who had lived for ≥ 6 months in Ansan (an urban area, n = 5,012) or Ansung (a rural area, n = 5,018) were recruited to the KoGES in 2001 and 2002. This prospective population-based study, which includes biennial examinations, is ongoing, with the latest follow-up done in 2021–2022. The study design and procedures have been described in a previous report [7]. Among 10,030 participants, 1,395 with missing data and 127 with symptomatic bacteriuria were excluded, leaving 8,508 who were recruited for this cross-sectional study.

### Outcome

Urine samples were collected as midstream clean-catch specimens and analyzed without centrifugation. Urinary bacteria were quantified using a UF-1000i flow

cytometer (Sysmex Corp., Kobe, Japan), which detects and counts particles based on fluorescence flow cytometry using a semiconductor laser.

The UF-1000i analyzes a defined volume of uncentrifuged urine and reports bacterial counts semi-quantitatively on a scale ranging from negative to 3+, according to the manufacturer's predefined thresholds.

In this study, we operationally defined bacteriuria without self-reported urgency–frequency symptoms—referred to as asymptomatic bacteriuria for the purposes of this epidemiologic analysis —as the presence of bacteriuria detected by urine flow cytometry (≥1+) in individuals who reported no urgency–frequency symptoms, based on a negative response to the question: "Have you recently experienced any sudden and frequent need to urinate?" This operational definition is based on flow cytometry rather than quantitative urine culture and therefore differs from the Infectious Diseases Society of America (IDSA) reference definition of asymptomatic bacteriuria [8,9].

## Measurements

Demographic information, medical history, family history, and lifestyle data were collected by trained interviewers using standardized questionnaires [7]. Blood pressure (BP) was measured in both arms after 5 min of rest, with the mean value of the two measurements used for the subsequent analyses. Weight was measured in kilograms to the nearest 0.1 kg, and height was measured within 0.1 cm without shoes. Body mass index (kg/m$^2$) was calculated by dividing each participant's weight (kg) by the square of their height (m$^2$), using the measurements acquired as described above. Fasting blood samples (approximately 20–30 mL) and urine samples (approximately 15 mL) were collected in the morning after at least 8 h of overnight fasting. Specimens were sent to a central laboratory (Seoul Clinical Laboratories, Seoul, Korea) for biochemical analysis. Total cholesterol, creatinine, blood urea nitrogen, fasting plasma glucose, and albumin were measured using an ADVIA 1650 chemistry analyzer (Bayer HealthCare Ltd., Tarrytown, NY, USA), and hemoglobin A1c was measured using high-performance liquid chromatography (Variant II; Bio-Rad Laboratories, Hercules, CA, USA).

## Definitions

Hypertension was defined as a physician-made diagnosis of hypertension, systolic BP ≥ 140 mmHg, diastolic BP ≥ 90 mmHg, or antihypertensive drug use. Diabetes was defined as a physician-made diagnosis of diabetes, fasting plasma glucose ≥ 7.0 mmol/L, 2-h glucose ≥ 11.1 mmol/L, hemoglobin A1c ≥ 6.5%, or the use of insulin or oral antidiabetic drugs. The estimated glomerular filtration rate was calculated using the Chronic Kidney Disease (CKD) Epidemiology Collaboration equation [10]. CKD was defined as an estimated glomerular filtration rate < 60 mL/min/1.73 m$^2$. Cardiovascular disease was defined as a physician-made diagnosis of angina, myocardial infarction, congestive heart failure, peripheral artery disease, or cerebrovascular disease. Age groups were categorized using a median value (50 years): older group (≥ 50 years) and younger group (< 50 years). The age cutoff of 50 years was chosen based on the median age of the study population and to facilitate interpretability in subgroup analyses, rather than to imply a biological threshold. As the KoGES cohort includes only adults aged 40–69 years, children and adolescents were not represented in this study. Serum albumin levels were categorized using a median value (42 g/L): high (≥ 42 g/L) and non-high (< 42 g/L). Physical activity was categorized as high (≥ 5 h per day) and non-high (< 5 h per day) based on exercise time, and included sports activities, hiking, running, logging, agricultural work, forestry, and mining. High monthly income was defined as earning ≥ 4 million Won. Self-rated health was evaluated through a participant-reported question: "Would you rate your general health as very poor, poor, fair, good, or excellent?" Responses of very poor or poor were categorized as poor self-rated health [11]. Impaired sleep was defined as a self-reported condition of insomnia, based on participant responses.

## Statistical analysis

Distributions of continuous variables were evaluated using histograms and Q-Q plots. The participants' C-reactive protein was not normally distributed. Normally distributed continuous variables are expressed as means ± standard deviations, non-normally distributed continuous variables as medians (interquartile ranges), and categorical variables as percentages. The odds ratios (ORs) and 95% confidence intervals (CIs) for ASB were determined using logistic regression analysis. Covariates were selected based on their clinical and statistical relevance. Only patients without missing values were included in the multivariate analysis. Multicollinearity among covariates was assessed using variance inflation factors (VIFs), with all VIF values below 2.2. Model adequacy was evaluated using the Hosmer–Lemeshow goodness-of-fit test, and no significant lack of fit was observed. All tests were two-sided, and a P value < 0.05 was considered statistically significant. All analyses were performed using R version 4.2.1 (R Foundation for Statistical Computing released 2021, Vienna, Austria).

## Results

The mean age of the 8,508 participants was 51.7 years, and the proportion of women was 51.4%. A total of 395 participants had ASB, with an overall prevalence of 4.6%. The prevalence of ASB in younger and older women was 5.9% and 9.5%, respectively, whereas that in younger and older men was 1.0% and 1.6%, respectively (Fig 1).

The baseline characteristics of participants based on their ASB status are presented in Table 1. People with ASB were older than those without ASB. The proportion of women was higher in the ASB group than that in the non-ASB group. As for the socioeconomic indicators, people with ASB showed lower proportions of current smoking and alcohol consumption than those without ASB. The ASB group also showed lower proportions of college graduates, high monthly income, and high physical activity, but higher proportions of poor self-rated health and impaired sleep than those in the non-ASB group. People with ASB showed higher body mass index and higher proportions of diabetes and CKD than those in the non-ASB group. The serum albumin and hemoglobin levels were lower in the ASB group than those in the non-ASB group.

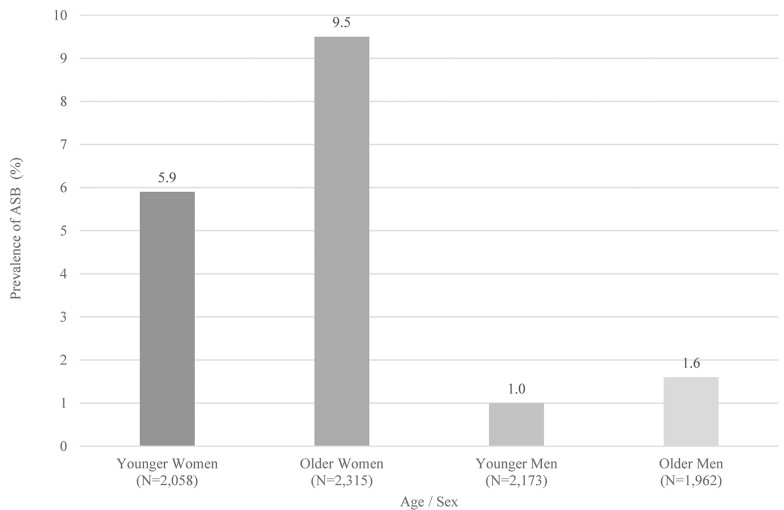

**Fig 1. Prevalence of asymptomatic bacteriuria (ASB) according to age group and sex.** Age groups were defined using the median age of 50 years. ASB was defined as bacteriuria detected by urine flow cytometry (≥1+) in the absence of self-reported urinary symptoms. Abbreviations: ASB, asymptomatic bacteriuria.

**Table 1. Clinical characteristics of the study population according to asymptomatic bacteriuria.**

|  | Non-ASB (N = 8,113) | ASB (N = 395) | P value |
|---|---|---|---|
| **Age (years)** | 51.6 ± 8.8 | 54.6 ± 9.2 | < 0.001 |
| **Women, n (%)** | 4,032 (49.7) | 341 (86.3) | < 0.001 |
| **Current smoking, n (%)** | 2,146 (26.5) | 40 (10.1) | < 0.001 |
| **Alcohol consumption, n (%)** | 4,013 (49.5) | 115 (29.1) | < 0.001 |
| **College graduate, n (%)** | 1,170 (14.4) | 26 (6.6) | < 0.001 |
| **High monthly income, n (%)** | 663 (8.2) | 17 (4.3) | 0.008 |
| **High physical activity, n (%)** | 1,350 (16.6) | 47 (11.9) | 0.016 |
| **Poor self-rated health, n (%)** | 2,594 (32.0) | 167 (42.3) | < 0.001 |
| **Impaired sleep, n (%)** | 1,275 (15.7) | 82 (20.8) | 0.009 |
| **Hypertension, n (%)** | 3,215 (39.6) | 173 (43.8) | 0.109 |
| **Diabetes, n (%)** | 1,001 (12.3) | 68 (17.2) | 0.005 |
| **CVD, n (%)** | 253 (3.1) | 17 (4.3) | 0.244 |
| **CKD, n (%)** | 172 (2.1) | 15 (3.8) | 0.04 |
| **Body mass index (kg/m²)** | 24.6 ± 3.1 | 25.0 ± 3.3 | 0.018 |
| **Fasting plasma glucose (mmol/L)** | 4.9 ± 1.2 | 4.7 ± 1.4 | 0.042 |
| **Serum Creatinine (μmol/L)** | 75.3 ± 17.5 | 68.6 ± 13.9 | < 0.001 |
| **Serum albumin (g/L)** | 42.7 ± 3.3 | 41.1 ± 2.7 | < 0.001 |
| **White blood cell (x10³/μL)** | 6.6 ± 1.8 | 6.4 ± 1.8 | 0.134 |
| **Cholesterol (mmol/L)** | 5.0 ± 0.9 | 4.9 ± 0.9 | 0.058 |
| **C-reactive protein (mg/L)** | 1.4 (0.7-2.5) | 1.4 (0.6-2.5) | 0.635 |
| **Hemoglobin (g/L)** | 137 ± 1.6 | 128 ± 1.3 | < 0.001 |

ASB, asymptomatic bacteriuria; CVD, cardiovascular disease; CKD, chronic kidney disease. Normally distributed continuous variables are expressed as the mean ± standard deviation, non-normally distributed continuous variables as the median (interquartile range), and categorical variables as percentages.

In the multivariate analysis (Table 2), older age, women, and diabetes were associated with increased odds of ASB, whereas high physical activity and serum albumin levels were associated with decreased odds of ASB. Current smoking, alcohol consumption, education and income levels, self-rated health, and sleep quality were not associated with ASB.

We combined four independent factors (age, diabetes status, physical activity, and serum albumin levels) with sex to analyze their association with ASB (Fig 2). Regardless of these independent factors, women showed higher odds of ASB than did men. Age and diabetes were not associated with ASB in men; however, older women (OR 1.608, 95% CI 1.256–2.066, P < 0.001) and women with diabetes (OR 1.579, 95% CI 1.144–2.148, P = 0.004) showed higher odds of ASB than did younger women and women without diabetes. Men with high physical activity (OR 0.332, 95% CI 0.114–0.766, P = 0.020) and serum albumin levels (OR 0.388, 95% CI 0.220–0.675, P = 0.001) showed decreased odds of ASB than did men without high physical activity and serum albumin levels. Women with high physical activity (OR 0.639, 95% CI 0.449-0.890, P = 0.010) and serum albumin levels (OR 0.684, 95% CI 0.538–0.866, P = 0.002) also showed decreased odds of ASB than did women without high physical activity and serum albumin levels.

## Discussion

Patients with ASB are commonly encountered in clinical practice. According to the 2019 recommendation statement by the U.S. Preventive Services Task Force, screening for and treatment of asymptomatic bacteriuria are not recommended in nonpregnant adults, as they provide no clinical benefit and may cause harm, whereas screening is recommended during

**Table 2. Factors associated with asymptomatic bacteriuria.**

| | Univariate OR (95% CI) | P value | Multivariate OR (95% CI) | P value |
|---|---|---|---|---|
| **Age (years)** | 1.038 (1.027-1.050) | < 0.001 | 1.030 (1.018-1.043) | < 0.001 |
| **Women** | 6.392 (4.825-8.634) | < 0.001 | 5.020 (3.307-7.714) | < 0.001 |
| **Current smoking** | 0.313 (0.222-0.430) | < 0.001 | 0.992 (0.662-1.462) | 0.970 |
| **Alcohol consumption** | 0.420 (0.335-0.522) | < 0.001 | 0.931 (0.725-1.189) | 0.572 |
| **College graduate** | 0.418 (0.273-0.612) | < 0.001 | 0.942 (0.603-1.415) | 0.782 |
| **High monthly income** | 0.505 (0.297-0.801) | 0.007 | 0.791 (0.459-1.275) | 0.365 |
| **High physical activity** | 0.677 (0.490-0.913) | 0.014 | 0.582 (0.417-0.795) | 0.001 |
| **Poor self-rated health** | 1.558 (1.268-1.911) | < 0.001 | 1.036 (0.831-1.290) | 0.750 |
| **Body mass index (kg/m²)** | 1.039 (1.006-1.072) | 0.018 | 1.000 (0.968-1.033) | 0.987 |
| **Impaired sleep** | 1.405 (1.088-1.796) | 0.008 | 0.952 (0.728-1.233) | 0.717 |
| **Hypertension** | 1.187 (0.968-1.455) | 0.099 | – | – |
| **Diabetes** | 1.477 (1.120-1.922) | 0.005 | 1.510 (1.125-2.001) | 0.005 |
| **CVD** | 1.397 (0.815-2.238) | 0.191 | – | – |
| **CKD** | 1.822 (1.021-3.016) | 0.029 | 1.333 (0.669-2.503) | 0.390 |
| **High serum albumin** | 0.836 (0.806-0.867) | < 0.001 | 0.882 (0.846-0.918) | < 0.001 |
| **Serum creatinine (µmol/L)** | 0.969 (0.961-0.976) | < 0.001 | 1.001 (0.992-1.007) | 0.858 |
| **White blood cell (x10³/µL)** | 0.957 (0.902-1.013) | 0.133 | – | – |
| **Cholesterol (mmol/L)** | 0.905 (0.809-1.011) | 0.079 | – | – |
| **C-reactive protein (mg/L)** | 0.995 (0.968-1.015) | 0.687 | – | – |
| **Hemoglobin (g/L)** | 0.733 (0.691-0.778) | < 0.001 | 0.995 (0.908-1.092) | 0.911 |

CVD, cardiovascular disease; CKD, chronic kidney disease; OR, odds ratio; CI, confidence interval. OR and 95% CI were calculated using logistic regression analysis. OR (95% CI) of (yes vs. no) for categorical variables and OR (95% CI) of per 1 unit increase for continuous variables were expressed. In multivariate analysis, only variables with P<0.05 were entered.

pregnancy [1]. However, the treatment rate remains high in most clinical practices [12], which may primarily be because symptoms related to urination are ambiguous, and ASB has been recognized as a risk factor for UTIs.

To reduce the overtreatment of ASB, it is necessary to distinguish ASB from UTIs; however, diagnostic tests are often difficult to perform in clinical settings, so they have largely been limited to smaller research studies. In this context, we conducted a large-scale study in a community-dwelling Korean adult population to investigate the prevalence and potential risk factors of ASB, using a pragmatic flow cytometry-based operational definition of bacteriuria rather than culture-based criteria.

The overall prevalence of ASB in this study was 4.6%: 5.9% in women under the age of 50, 9.5% in women over 50, 1.0% in men under 50, and 1.6% in men over 50. These results aligned with previous studies, which reported 2–4.8% in women under the age of 50, 2.8–8.6% in women over 50, and 0.6–1.5% in men over 50 [1].

We confirmed that older age, female sex, and diabetes were independently associated with ASB, which has been consistently reported in previous studies [1]. In agreement with others, we assumed that the female sex had the strongest association with ASB prevalence. In the current study, people with ASB were mostly women (86.3%). The pattern of socioeconomic status in the ASB group was similar to that of Korean women in the early 2000s. This may explain why most socioeconomic indicators (smoking and alcohol consumption, education and income status, self-rated health, and sleep quality) lost statistical significance in the multivariate analysis. Anatomically, the urethra in women is short and close to the anus; therefore, women are generally more prone to bacterial exposure in this region. Changes in vaginal conditions following menopause can also promote bacterial colonization [13]. However, parity could not be considered due to data unavailability, which may have led to residual confounding in the association between female sex and asymptomatic

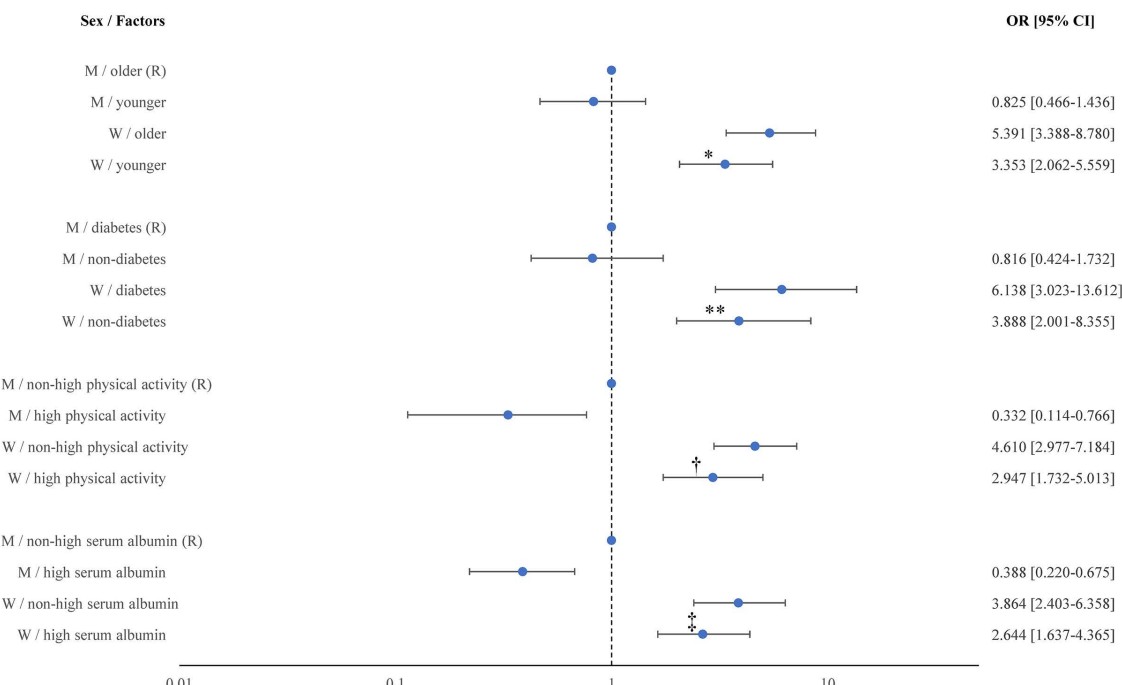

**Fig 2. Odds ratios (ORs) for asymptomatic bacteriuria according to sex and selected factors.** ORs and 95% confidence intervals (CIs) were obtained from multivariate logistic regression models adjusted for age, sex, alcohol consumption, smoking status, education level, physical activity, monthly income, self-rated health, impaired sleep, diabetes, chronic kidney disease, body mass index, serum creatinine, serum albumin, and hemoglobin. When a variable was used for stratification, it was excluded from the corresponding regression model. Statistical significance was assessed using two-sided Wald tests. P values are indicated by symbols as follows: P<0.001 (*), P=0.004 (**), P=0.010 (†), and P=0.002 (‡). Abbreviations: OR, odds ratio; CI, confidence interval; W, women; M, men. R, reference.

bacteriuria. In addition to sex-related factors, reduced bladder mobility and urination-related disorders, such as urinary incontinence, in older adults [14], as well as the harmful effects of hyperglycemia on immune function [15,16] and bladder function in diabetes [17] may also contribute to the increased prevalence of ASB.

We found that physical activity and serum albumin were independently associated with ASB, with higher levels of both factors being related to lower odds of ASB. The inverse association between physical activity and ASB is biologically plausible and may be explained by several potential mechanisms, as regular physical activity has been associated with improved insulin sensitivity and glycemic control [18], enhanced innate and adaptive immune responses [19], and improved lower urinary tract function, which could help reduce urinary stasis and [20], consequently, susceptibility to bacterial colonization. Serum albumin reflects not only nutritional status but also systemic inflammation and immune function, as it is a negative acute-phase reactant whose synthesis decreases during inflammatory states, and lower serum albumin levels have been associated with impaired host defense mechanisms and increased susceptibility to infection [21,22]. From an epidemiological perspective, serum albumin is a readily available clinical marker, and together with physical activity, these modifiable lifestyle and nutritional factors may be relevant to ASB risk beyond fixed demographic characteristics and could represent potential targets for population-level prevention strategies.

This study has several notable strengths and limitations. First, the cross-sectional design limits the ability to draw definitive conclusions about causality and increases the potential for bias or confounding effects. However, we employed an appropriate methodology to examine the prevalence and potential risk factors of ASB in a large population. Further studies are needed to clarify causal relationships and explore the impact of various factors in greater depth. Second, bacteriuria

without self-reported urgency–frequency symptoms was not defined using urine culture, the diagnostic gold standard, but rather by urine flow cytometry combined with the absence of self-reported urinary symptoms. Although this approach may lead to some misclassification, particularly in cases of low-grade bacteriuria or unrecognized symptoms, urine flow cytometry correlates well with conventional culture results and demonstrates high sensitivity for clinically significant bacteriuria. As such, this pragmatic approach enabled the evaluation of bacteriuria without self-reported urgency–frequency symptoms in a large community-dwelling population. Moreover, participants with missing data who were excluded from the analysis had a substantially higher prevalence of diabetes than those included in the final analytic cohort. Because diabetes was independently associated with increased odds of ASB, the exclusion of this higher-risk subgroup may have resulted in a slight underestimation of the true prevalence of ASB in the general population. Third, symptom-based exclusion was conducted using a single self-reported question on urgency–frequency symptoms, as no additional urinary or systemic symptom variables were available in the dataset. Consequently, individuals with other urinary or systemic symptoms may have been misclassified as asymptomatic, while some participants with non-infectious causes of urinary frequency may have been excluded. This limitation reflects the constraints of the available data rather than the symptom screening strategy itself, and future studies incorporating a broader range of urinary and systemic symptoms are warranted to more accurately characterize bacteriuria in the general population. Lastly, only participants of Asian descent were included. Therefore, the results may be difficult to generalize among other ethnicities; nevertheless, they present the characteristics of ASB in Asian adults, which has not been studied in this demographic.

In conclusion, we investigated the prevalence of ASB and identified potential risk factors in a large community of Korean adults. In addition to the previously recognized factors of age, sex, and diabetes, we identified physical activity and serum albumin levels as novel, potentially modifiable factors that may be associated with a reduced likelihood of ASB and future UTIs.

## Acknowledgments

Data in this study were from the Korean Genome and Epidemiology Study (KoGES;6635−302), National Institute of Health, Korea Disease Control and Prevention Agency, Republic of Korea.

## Author contributions

**Conceptualization:** Sung Woo Lee.

**Formal analysis:** Hyun Lee Ko, Soojin Lee, Dha Woon Im, Sung Woo Lee.

**Project administration:** Sung Woo Lee.

**Supervision:** Sung Woo Lee.

**Writing – original draft:** Hyun Lee Ko.

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
