## [Decision Letter · Decision Letter 0]

10 Dec 2025

Dear Dr. ko,

Thank you for submitting your manuscript to PLOS ONE. After careful consideration, we feel that it has merit but does not fully meet PLOS ONE’s publication criteria as it currently stands. Therefore, we invite you to submit a revised version of the manuscript that addresses the points raised during the review process.

**Summary of Major Revisions**

**1. Strengthened Methodology and Clarity of ASB Definition (Reviewer 5, Reviewer 3)**

We have revised the definition of asymptomatic bacteriuria (ASB) to more transparently reflect that our operational classification is based on **flow cytometry (UF-1000i ≥1+) and absence of self-reported urinary symptoms** , acknowledging that this differs from the IDSA gold standard definition.We added citations validating flow cytometry thresholds and expanded the limitations around possible misclassification.

**2. Expanded Description of Flow Cytometry Procedures (Reviewer 1, Reviewer 5)**

Additional details regarding urine sample handling, UF-1000i measurement parameters, volume analyzed, and potential false-positive and false-negative considerations were added to the Methods section.

**3. Improved Statistical Rigor and Reporting (Reviewer 5, Reviewer 3, Reviewer 1)**

Reviewer 5’s concerns regarding model specification, variable cut-points, collinearity, and covariate selection have been addressed by:Clearly defining our variable selection strategy.Providing justification for the chosen cut-points and evaluating models using continuous variables as sensitivity checks.Adding a description of multicollinearity assessment (VIF values).Expanding model diagnostics (goodness-of-fit).In response to Reviewer 1, we added explicit statistical analysis details in Figure 2 and clarified significance testing procedures.**4. Handling of Missing Data and Participant Flow (Reviewer 5)**We now provide a detailed **participant flow diagram** , specify the reasons for excluding 1,395 individuals, and discuss the potential impact of missing data.Where applicable, we conducted sensitivity analyses and clarified why multiple imputation was not applied.**5. Updated and Expanded Literature Review (Reviewer 1, Reviewer 2, Reviewer 5)**To address the comments about outdated references (Reviewer 2) and weak discussion depth (Reviewer 1 and Reviewer 5), we:Updated references 1–3, 5, 9, 13, and 16–25 with recent post-2019 studies and guidelines.Expanded the final paragraphs of the Discussion with appropriate citations.Added contemporary epidemiological and methodological literature to contextualize our results.6. Expanded Interpretation and Novel Insights (Reviewer 3, Reviewer 5)We incorporated reviewer suggestions to explore potential interaction effects (e.g., sex × physical activity) and report findings where meaningful.We enriched the discussion on the protective associations of serum albumin and physical activity, linking them to emerging mechanistic literature.

**7. Addressing Demographic Clarifications (Reviewer 4)**

We clarified the rationale behind using two age categories and acknowledged the limitations related to excluding children and adolescents.For female participants, we explicitly state the unavailability of parity data in KoGES and discuss its potential relevance as a limitation.

**8. Data Availability Compliance (Reviewer 1, Reviewer 5)**

To align with PLOS ONE’s data-sharing policy, we revised the Data Availability Statement to explicitly describe access through the **KoGES/KDCA repository mechanism** , replacing any implication of author-controlled access.

**9. Figure Improvements and Standardization (Reviewer 2, Reviewer 5)**

Appropriate footnotes have been added to Figures 1 and 2.Decimal formatting, unit consistency, and abbreviation definitions have been fully standardized across the manuscript.

**10. General Language and Presentation Improvements (All reviewers)**

Although reviewers found the language generally clear, we conducted an additional round of professional editing to ensure clarity, coherence, and precision.

We look forward to receiving your revised manuscript.

Kind regards,

Hari Murthy, Ph.D.

Academic Editor

PLOS One

“EMBRI Grants 2024-EMBRIUI-0004 from the Eulji University.”

“Data in this study were from the Korean Genome and Epidemiology Study (KoGES;6635-302), National Institute of Health, Korea Disease Control and Prevention Agency, Republic of Korea. This research was supported by EMBRI Grants 2024-EMBRIUI-0004 from the Eulji University.”

“EMBRI Grants 2024-EMBRIUI-0004 from the Eulji University.”

4. In the online submission form you indicate that your data is not available for proprietary reasons and have provided a contact point for accessing this data. Please note that your current contact point is a co-author on this manuscript. According to our Data Policy, the contact point must not be an author on the manuscript and must be an institutional contact, ideally not an individual. Please revise your data statement to a non-author institutional point of contact, such as a data access or ethics committee, and send this to us via return email. Please also include contact information for the third party organization, and please include the full citation of where the data can be found.

Reviewers' comments:

Reviewer's Responses to Questions

**Comments to the Author**

1. Is the manuscript technically sound, and do the data support the conclusions?

Reviewer #1: Yes

Reviewer #2: Yes

Reviewer #3: Yes

Reviewer #4: Yes

Reviewer #5: Partly

2. Has the statistical analysis been performed appropriately and rigorously?

Reviewer #1: Yes

Reviewer #2: Yes

Reviewer #3: Yes

Reviewer #4: Yes

Reviewer #5: No

3. Have the authors made all data underlying the findings in their manuscript fully available?

Reviewer #1: No

Reviewer #2: Yes

Reviewer #3: Yes

Reviewer #4: Yes

Reviewer #5: No

4. Is the manuscript presented in an intelligible fashion and written in standard English?

Reviewer #1: Yes

Reviewer #2: Yes

Reviewer #3: Yes

Reviewer #4: Yes

Reviewer #5: Yes

Reviewer #1: The authors provided data regarding the prevalence and risk factors of asymptomatic bacteriuria in community-dwelling Korean adults. However, I have a comment on the weak discussion section, as no single reference was cited in the last paragraphs.

In Fig. 2, can the authors provide the statistical analysis used to determine significance?.

The authors need to provide information about the false positives and negative results of flow cytometry.

More details regarding the flow cytometer analysis should be mentioned in the Materials and Methods section.

Reviewer #2: Dear author,

The article titled "Prevalence and risk factors of asymptomatic bacteriuria in community-dwelling Korean adults" was reviewed. Please make the following corrections:

1- References 1-3, 5, 9, 13, and 16-25 are old, please use new references.

2- Please insert the suitable footnotes for Figures 1 and 2.

Kind regards

Reviewer #3: This manuscript presents a large-scale, population-based analysis of asymptomatic bacteriuria (ASB) in community-dwelling Korean adults, using data from the Korean Genome Epidemiology Study (KoGES). The study is technically sound, and the data support the conclusions drawn. Below are detailed comments addressing the key review criteria:

-The study design is robust, leveraging a large sample size (n = 8,508) and excluding participants with missing data or symptomatic bacteriuria, which strengthens the internal validity.

-ASB is clearly defined using both flow cytometry results and self-reported symptoms, aligning with accepted clinical standards.

-The use of flow cytometry for bacteriuria detection is appropriate for large-scale screening and is supported by prior literature.

-The conclusions regarding associations between ASB and factors such as age, sex, diabetes, physical activity, and serum albumin levels are well-supported by the data and statistical analysis.

-The study adheres to ethical standards, with approval from the relevant institutional review board and appropriate handling of anonymized data.

-The authors have declared no competing interests, and funding sources are transparently disclosed.

-The data availability statement is clear and appropriately addresses privacy concerns.

-The manuscript would benefit from a brief discussion of model diagnostics (e.g., goodness-of-fit, multicollinearity), although this omission does not undermine the validity of the findings.

-Suggestions for Improvement:

-Explore whether interaction effects (e.g., between sex and physical activity) might yield additional insights.

--This is a well-executed and valuable contribution to the literature on ASB in Asian populations. The findings are novel, particularly the protective associations of physical activity and serum albumin, and warrant further investigation in future longitudinal studies.

Reviewer #4: Overall, the study provided a comperhensive data analysis for the cohort included in the study. However, some questions need to be clrified:

1- The authors devided the study subjects into two age groups only, and named age under 50 yr as young!! while exluded the younger ages (childhood and adilescence) age groups.

2- Female subjects lack some information like the number of parturition, which could be an important parameter to lock at in the analysis.

Reviewer #5: This manuscript examines the prevalence and risk factors of asymptomatic bacteriuria (ASB) among 8,508 community-dwelling Korean adults enrolled in the KoGES cohort. The authors employ standardized biochemical and clinical measurements and logistic regression analyses to identify factors associated with ASB, including age, sex, diabetes, serum albumin, and physical activity. The topic is clinically relevant and timely, as large-scale, population-based data on ASB in Asian adults remain scarce. However, several methodological and interpretive limitations limit the robustness of the conclusions.

Major Strengths

• Large, well-characterized cohort: The KoGES sample provides a robust foundation with standardized data and broad demographic coverage.

• Clear research objective: The study targets an under-investigated population (Asian adults) with implications for clinical screening and antimicrobial stewardship.

• Sound basic analysis: Logistic regression modeling and transparent presentation of odds ratios enhance clarity and reproducibility.

• Ethical and procedural compliance: Ethical approval and participant consent are clearly stated.

• Novel associations: The inverse relationships between ASB and both serum albumin and physical activity merit attention for future mechanistic research.

Major Concerns (Revisions Required)

• Definition and ascertainment of ASB: ASB is defined using flow cytometry (UF-1000i, ≥1+) rather than quantitative culture, and asymptomatic status is based on a single symptom question. This diverges from IDSA criteria and risks misclassification. Action: Re-frame as 'bacteriuria without self-reported urgency-frequency symptoms', cite validation for the ≥1+ threshold, and emphasize this limitation.

• Incomplete exclusion of symptomatic cases: Only one urinary symptom was assessed. Clarify all symptom questions available and discuss the likelihood of misclassification bias.

• Arbitrary cut-points: Thresholds for age, activity, and albumin appear data-driven. Explain or justify these, or treat variables continuously.

• Handling of missing data: Approx. 1,395 participants were excluded but missingness is not described. Provide a flow diagram and justify the approach or use multiple imputation.

• Model specification and confounding: Covariate selection strategy unclear; assess potential collinearity. Clarify modeling procedure and temper causal interpretations.

• Discussion depth and literature currency: Relies heavily on older studies. Update with post-2019 ASB guidelines and literature to contextualize findings.

• Data availability statement: Restricting access to corresponding author conflicts with PLOS ONE policy. Specify formal KoGES/KDCA mechanism and justify any restrictions.

Minor Comments

Figure 1: Caption should read: 'Prevalence of asymptomatic bacteriuria according to age and sex.'

Methods: Specify urine volume analyzed and sample handling (for UF-1000i and serum assays).

Results: Define all abbreviations at first use (ASB, CKD, CVD, BMI).

Formatting: Standardize decimal points (e.g., '0.995' rather than '0,995') and units; ensure consistency.

References: Update with more recent ASB epidemiology and methodological sources.

Generally, the manuscript addresses an important epidemiologic question with a valuable dataset, but substantial clarification and methodological transparency are required before the conclusions can be considered reliable. Addressing definitional, analytic, and interpretive issues will strengthen the paper.

**Do you want your identity to be public for this peer review?** For information about this choice, including consent withdrawal, please see our Privacy Policy

Reviewer #1: No

Reviewer #2: No

Reviewer #3: **Yes:** Ismail Yosri Abdelgelel

Reviewer #4: No

Reviewer #5: No

---

## [Author Response · Author response to Decision Letter 1]

13 Jan 2026

[Comment 1] Definition of ASB

1. Response to the reviewer

Thank you for this important comment. We acknowledge that our operational definition differs from the standard IDSA/USPSTF criteria and have now explicitly stated this in the manuscript, along with justification for using urine flow cytometry–based bacteriuria (≥1+) and appropriate supporting references.

2. Location of changes in the manuscript

Methods – Outcome (Page 4, Lines 84–90)

Methods – Outcome (Page 5, Lines 91–98)

Methods – Measurements (Page 5, Lines 106–108)

Discussion – Limitations (Page 14-15, Lines 257–264)

3. Revised text

• Urine samples were collected as midstream clean-catch specimens and analyzed without centrifugation. Urinary bacteria were quantified using a UF-1000i flow cytometer (Sysmex Corp., Kobe, Japan), which detects and counts particles based on fluorescence flow cytometry using a semiconductor laser.

The UF-1000i analyzes a defined volume of uncentrifuged urine and reports bacterial counts semi-quantitatively on a scale ranging from negative to 3+, according to the manufacturer’s predefined thresholds.

• In this study, we operationally defined bacteriuria without self-reported urgency–frequency symptoms—referred to as asymptomatic bacteriuria for the purposes of this epidemiologic analysis—as the presence of bacteriuria detected by urine flow cytometry (≥1+)* in individuals who reported no urgency–frequency symptoms, based on a negative response to the question: “Have you recently experienced any sudden and frequent need to urinate?” This operational definition is based on flow cytometry rather than quantitative urine culture and therefore differs from the Infectious Diseases Society of America (IDSA) reference definition of asymptomatic bacteriuria.

*Reference

1.Validation and Search of the Ideal Cut-Off of the Sysmex UF-1000i® Flow Cytometer for the Diagnosis of Urinary Tract Infection in a Tertiary Hospital in Spain. Front Med (Lausanne). 2018 Apr 9;5:92.

2. Diagnosis of Bacteriuria and Leukocyturia by Automated Flow Cytometry Compared with Urine Culture. J Clin Microbiol. 2010 Aug 25;48(11):3990–3996.

• Fasting blood samples (approximately 20–30 mL) and urine samples (approximately 15 mL) were collected in the morning after at least 8 h of overnight fasting.

• Second, bacteriuria without self-reported urgency–frequency symptoms was not defined using urine culture, the diagnostic gold standard, but rather by urine flow cytometry combined with the absence of self-reported urinary symptoms. Although this approach may lead to some misclassification, particularly in cases of low-grade bacteriuria or unrecognized symptoms, urine flow cytometry correlates well with conventional culture results and demonstrates high sensitivity for clinically significant bacteriuria. As such, this pragmatic approach enabled the evaluation of bacteriuria without self-reported urgency–frequency symptoms in a large community-dwelling population.

[Comment 2] Symptom Ascertainment

1. Response to the reviewer

We appreciate the reviewer’s concern regarding symptom ascertainment based on a single question. Because the provided dataset did not include urinary or systemic symptoms other than self-reported urgency–frequency, we were unable to further exclude individuals with potential urinary tract infection–related symptoms beyond this single item. This limitation reflects constraints of the available data rather than the study design itself. Accordingly, we have revised the Limitations section to explicitly acknowledge the potential for misclassification arising from the restricted symptom assessment.

2. Location of changes in the manuscript

Discussion – Limitations (Page 15, Lines 264–272)

3. Revised text

Third, symptom-based exclusion was conducted using a single self-reported question on urgency–frequency symptoms, as no additional urinary or systemic symptom variables were available in the dataset. Consequently, individuals with other urinary or systemic symptoms may have been misclassified as asymptomatic, while some participants with non-infectious causes of urinary frequency may have been excluded. This limitation reflects the constraints of the available data rather than the symptom screening strategy itself, and future studies incorporating a broader range of urinary and systemic symptoms are warranted to more accurately characterize bacteriuria in the general population.

[Comment 3] Categorization of Variables

1. Response to the reviewer

We thank the reviewer for raising concerns about variable categorization. While it was not possible to provide external justification for all cutoffs, age and sex were categorized using median values to reflect clinically meaningful groupings; in particular, the age cutoff corresponds to a period of hormonal and physiological changes in women that may influence bacteriuria epidemiology. This rationale has been clarified in the manuscript. For other variables, including serum albumin and physical activity, we conducted sensitivity analyses using continuous forms and alternative categorizations, which yielded results consistent with the primary analyses.

• Age was categorized using a cut-off of 50 years, corresponding approximately to the median age of the study population and reflecting a clinically relevant threshold associated with hormonal and physiological changes, particularly in women, which have been reported to influence the epidemiology of bacteriuria. This categorization was intended to facilitate interpretability and subgroup comparisons rather than to imply a biological dichotomy.

Reference

Raz R; Urinary tract infection in postmenopausal women.; Korean Journal of Urology. 2011;52(12):801–808.

• Covariates included in the multivariate logistic regression models were selected based on prior clinical relevance and variables showing statistical significance in univariate analyses. Continuous variables were categorized using median values to facilitate clinical interpretability and subgroup comparisons. To assess the robustness of our findings, sensitivity analyses were additionally performed using continuous forms of these variables, yielding consistent results.

Variable Sensitivity analysis (continuous)

OR (95% CI)

Age (years) 1.061 (0.890-1.264)

Albumin (g/L) 0.917 (0.723-1.159)

Age * albumin 0.999 (0.995-1.004)

Physical activity (alternative categorical)

1 (less than 30 minutes) 0.737 (0.403-1.240)

2 (30 to less than 60 minutes) 1.140 (0.685-1.801)

3 (60 to less than 90 minutes) 0.757 (0.353-1.425)

4 (90 minutes to less than 2 hours) 1.378 (0.685-2.512)

5 (2 hours to less than 3 hours) 1.117 (0.619-1.877)

6 (3 hours to less than 4 hours) 0.836 (0.405-1.543)

7 (4 hours to less than 5 hours) 0.560 (0.233-1.138)

8 (5 hours or more) 0.567 (0.404-0.781)

[Comment 4] Handling of missing data

1. Response to the reviewer

We appreciate this comment regarding missing data. In response, we added a participant flow diagram and compared baseline characteristics between participants included in the final analysis and those excluded due to missing data to assess potential selection bias.

• Participant flow diagram

• Comparison table of included and excluded participants

variables Included participants (N = 8,508) Excluded participants

(N = 1,395) P value

Age (years) 51.7 ± 8.8 55.5 ± 8.8 < 0.001

Men (%) 4,135 (48.6) 610 (43.7) 0.001

Current smoking (%) 2,186 (25.7) 351 (27.8) 0.113

Alcohol consumption (%) 4,128 (48.5) 520 (39.9) < 0.001

Diabetes (%) 1,069 (12.6) 359 (44.6) < 0.001

Chronic kidney disease (%) 187 (2.2) 31 (2.2) 1.000

Body mass index (kg/m2) 24.6 ± 3.1 24.4 ± 3.2 0.007

Serum creatinine (µmol/L) 75.0 ± 17.4 72.1 ± 33.2 0.001

Serum albumin (g/L) 42.6 ± 3.3 40.8 ± 2.5 < 0.001

[Comment 5] Confounding and model specification

1. Response to the reviewer

Thank you for this methodological comment. We have clarified our covariate selection strategy based on clinical relevance and prior literature, and assessed multicollinearity among covariates using variance inflation factors (VIFs), with all VIF values below 2.2, indicating no significant multicollinearity. Model adequacy was evaluated using goodness-of-fit measures, including the Hosmer–Lemeshow test, and no significant lack of fit was observed (Hosmer–Lemeshow χ² = 5.10, df = 8, P = 0.75). All statistical tests were two-sided, and a P value < 0.05 was considered statistically significant. Nevertheless, residual confounding due to unmeasured or incompletely measured factors cannot be entirely ruled out.

2. Location of changes in the manuscript

Statistical Analysis (Page 7, Lines 145–148)

3. Revised text

Multicollinearity among covariates was assessed using variance inflation factors (VIFs), with all VIF values below 2.2. Model adequacy was evaluated using the Hosmer–Lemeshow goodness-of-fit test, and no significant lack of fit was observed. All tests were two-sided, and a P value < 0.05 was considered statistically significant.

Variables VIF

Age 1.23

Sex 2.11

Current smoking 1.39

Alcohol consumption 1.20

College graduate 1.04

High monthly income 1.08

High physical activity 1.03

Poor self-rated health 1.10

Impaired sleep 1.06

Diabetes 1.07

Chronic kidney disease 1.41

Body mass index 1.09

Serum creatinine 1.57

Serum albumin 1.10

Hemoglobin 1.51

[Comment 6] Depth and currency of literature

1. Response to the reviewer

We appreciate the reviewer’s suggestion to strengthen the depth and currency of the literature. In response, we revised the Discussion to incorporate contemporary clinical guidelines and recent evidence on asymptomatic bacteriuria, including updated recommendations from major organizations. In addition, several older references were replaced or supplemented with more recent studies to better reflect current understanding and clinical practice.

2. Location of changes in the manuscript

Discussion (Page 12, Lines 206–210)

Discussion (Page 14, Lines 237–251)

References list (updated to reflect revised citations)

3. Revised text

• Patients with ASB are commonly encountered in clinical practice. According to the 2019 recommendation statement by the U.S. Preventive Services Task Force, screening for and treatment of asymptomatic bacteriuria are not recommended in nonpregnant adults, as they provide no clinical benefit and may cause harm, whereas screening is recommended during pregnancy (1).

Reference

U.S. Preventive Services Task Force Owens DK, Davidson KW, Krist AH, et al.

Screening for Asymptomatic Bacteriuria in Adults: US Preventive Services Task Force Recommendation Statement. JAMA. 2019;322(12):1188–1194.

• We found that physical activity and serum albumin were independently associated with ASB, with higher levels of both factors being related to lower odds of ASB. The inverse association between physical activity and ASB is biologically plausible and may be explained by several potential mechanisms, as regular physical activity has been associated with improved insulin sensitivity and glycemic control, enhanced innate and adaptive immune responses [], and improved lower urinary tract function, which could help reduce urinary stasis and, consequently, susceptibility to bacterial colonization []. Serum albumin reflects not only nutritional status but also systemic inflammation and immune function, as it is a negative acute-phase reactant whose synthesis decreases during inflammatory states, and lower serum albumin levels have been associated with impaired host defense mechanisms and increased susceptibility to infection []. From an epidemiological perspective, serum albumin is a readily available clinical marker, and together with physical activity, these modifiable lifestyle and nutritional factors may be relevant to ASB risk beyond fixed demographic characteristics and could represent potential targets for population-level prevention strategies.

[Comment 7] Data availability statement

1. Response to the reviewer

Thank you for this comment. We revised the Data Availability Statement to comply with PLOS ONE policy.

2. Location

Changes were made to the Data Availability Statement file.

3. Revised text

The data underlying the results of this study are available from the Korean Genome and Epidemiology Study (KoGES), National Institute of Health. Due to ethical and legal restrictions, the data are not publicly available but can be accessed by qualified researchers through an application process administered by the National Institute of Health (http://coda.nih.go.kr). The authors did not have special access privileges and any researcher meeting the access criteria may obtain the data in the same manner.

[Comment 8] Presentation and Clarity

1. Response to the reviewer

We thank the reviewer for the helpful comments regarding presentation and clarity. In response, we carefully reviewed the manuscript to address minor typographical errors, improve clarity, and ensure consistency throughout the text. Specifically, figure captions and footnotes were revised, abbreviations were defined at first use, numerical formats were standardized, and all laboratory variables were expressed using consistent SI units.

2. Location of changes in the manuscript

Figure legends and footnotes (Page 7, Lines 156–159 & Page 12, Lines 196-204)

Methods – Definitions (Page 6, Lines 125–128)

Discussion (Page 13, Lines 231–233)

Minor editorial and formatting issues, including unit consistency, spacing, punctuation, and abbreviation usage, were corrected throughout the manuscript (Methods, Results, Tables, and Figure legends).

3. Revised text

• Figure 1. Prevalence of asymptomatic bacteriuria (ASB) according to age group and sex. Age groups were defined using the median age of 50 years. ASB was defined as bacteriuria detected by urine flow cytometry (≥1+) in the absence of self-reported urinary symptoms. Abbreviations: ASB, asymptomatic bacteriuria.

• Figure 2. Odds ratios (ORs) for asymptomatic bacteriuria according to sex and selected factors. ORs and 95% confidence intervals (CIs) were obtained from multivariate logistic regression models adjusted for age, sex, alcohol consumption, smoking status, education level, physical activity, monthly income, self-rated health, impaired sleep, diabetes, chronic kidney disease, body mass index, serum creatinine, serum albumin, and hemoglobin. When a variable was used for stratification, it was excluded from the corresponding regression model. Statistical significance was assessed using two-sided Wald tests. P values are indicated by symbols as follows: P < 0.001 (*), P = 0.004 (**), P = 0.010 (†), and P = 0.002 (‡). Abbreviations: OR, odds ratio; CI, confidence interval; W, women; M, men.

• The age cutoff of 50 years was chosen based on the median age of the study population and to facilitate interpretability in subgroup analyses, rather than to imply a biological threshold. As the KoGES cohort includes only adults aged 40–69 years, children and adolescents were not represented in this study.

• However, parity could not be considered due to data unavailability, which may have led to residual confounding in the association between female sex and asymptomatic bacteriuria.

---

## [Decision Letter · Decision Letter 1]

24 Feb 2026

Prevalence and risk factors of asymptomatic bacteriuria in community-dwelling Korean adults

PONE-D-25-33638R1

Dear Dr. ko,

We’re pleased to inform you that your manuscript has been judged scientifically suitable for publication and will be formally accepted for publication once it meets all outstanding technical requirements.

Kind regards,

Hari Murthy, Ph.D.

Academic Editor

PLOS One

Additional Editor Comments (optional):

The reviewers have provided their comments and based on their comments, there are few minor grammar issues and including recent references in the bibliography. Request you to do a thorough proof reading and update the reference.

Reviewers' comments:

Reviewer's Responses to Questions

**Comments to the Author**

Reviewer #2: (No Response)

Reviewer #3: All comments have been addressed

Reviewer #4: All comments have been addressed

2. Is the manuscript technically sound, and do the data support the conclusions?

Reviewer #2: Yes

Reviewer #3: Yes

Reviewer #4: Yes

3. Has the statistical analysis been performed appropriately and rigorously?

Reviewer #2: Yes

Reviewer #3: Yes

Reviewer #4: Yes

4. Have the authors made all data underlying the findings in their manuscript fully available?

Reviewer #2: Yes

Reviewer #3: No

Reviewer #4: Yes

5. Is the manuscript presented in an intelligible fashion and written in standard English?

Reviewer #2: Yes

Reviewer #3: Yes

Reviewer #4: Yes

Reviewer #2: Dear author,

The article titled "Prevalence and risk factors of asymptomatic bacteriuria in community-dwelling Korean adults" was reviewed. It provides useful information for readers. Please make the following correction:

References 6, 9, 14, 16, 17, 20, and 21 are old, please use new references.

Kind regards

Reviewer #3: The manuscript has been significantly strengthened by the inclusion of a participant flow diagram, a comparison table for excluded participants, and a more transparent discussion regarding the "operational definition" of asymptomatic bacteriuria (ASB) used in this study.

The study's large sample size (N=8,508) remains its primary strength, allowing for meaningful epidemiological analysis despite the limitations inherent in retrospective data.The identification of high physical activity and serum albumin levels as independent protective factors is a novel and scientifically valuable finding that warrants publication.

I recommend the manuscript for publication, provided the authors address the following minor technical errors found in the revised text.

1-I appreciate the authors' honesty in redefining their outcome as an "operational definition" based on flow cytometry (≥1+) rather than the IDSA culture-based standard.The added text in the Methods and Limitations sections adequately contextualizes this choice for the reader. This pragmatic approach is acceptable for a large-scale population study where culture is unavailable, provided the distinction remains clear throughout the text.

2-The addition of the participant flow diagram and the comparison table for missing data is excellent.The analysis reveals that the excluded group had a significantly higher prevalence of diabetes (44.6% vs. 12.6%). While the authors have noted this, it is worth ensuring the Discussion briefly acknowledges that the exclusion of this high-risk diabetic subgroup might mean the true prevalence of ASB in the general population could be slightly higher than reported.

Reviewer #4: (No Response)

**Do you want your identity to be public for this peer review?** For information about this choice, including consent withdrawal, please see our Privacy Policy

Reviewer #2: No

Reviewer #3: **Yes:** Ismail Yosri Abdelgelel Ismail

Reviewer #4: **Yes:** Ali Hadi Abbas

---

## [Editor Report · Acceptance letter]

PONE-D-25-33638R1

PLOS One

Dear Dr. ko,

I'm pleased to inform you that your manuscript has been deemed suitable for publication in PLOS One. Congratulations! Your manuscript is now being handed over to our production team.

Kind regards,

on behalf of

Dr. Hari Murthy

Academic Editor

PLOS One